# Brazing of Copper Pipes for Heat Pump and Refrigeration Applications

António B. Pereira [1,2], João M. S. Dias [1,2,*], José P. Rios [1], Nélia M. Silva [3], Sathishkumar Duraisamy [1] and Ana Horovistiz [1,2]

1. TEMA—Centre for Mechanical Technology and Automation, Department of Mechanical Engineering, Campus de Santiago, University of Aveiro, 3810-193 Aveiro, Portugal; abastos@ua.pt (A.B.P.); horovistiz@ua.pt (A.H.)
2. LASI—Intelligent Systems Associate Laboratory, 4800-058 Guimarães, Portugal
3. CIDMA—Centre for Research & Development in Mathematics and Applications, Department of Mathematics, Campus de Santiago, University of Aveiro, 3810-193 Aveiro, Portugal
* Correspondence: joaomdias@ua.pt

**Abstract:** In heat pumps and refrigeration systems, copper parts play a crucial role. Since heat pumps for space and water heating work under high pressure and are susceptible to vibrations, it is crucial to perfectly weld the copper pipes and heat exchangers to avoid system failures and prevent the leakage of the circulating refrigerants, which are harmful to the environment. The welding of the copper pipes is usually performed by the brazing process in a furnace. The components are subjected to a period of approximately 50 min inside a continuously open oven, varying the temperature from 710 °C to 830 °C. The oven inlets and outlets are protected by nitrogen curtains to guarantee a suitable internal environment and prevent the contamination of the gas inside the oven. This work analyses which welding methods are most suitable for welding copper, the best joint shape, process time, brazing specimens of a copper alloy, tightness tests, and mechanical properties and composition of the welding samples. From the tests carried out, the appearance of small and large defects is reduced by using a 1 mm thick external ring of filler material and a brazing temperature of 820 °C.

**Keywords:** brazing; copper pipe; heat pumps



## 1. Introduction

Heat pumps have been identified by the International Energy Agency (IEA) as the main technology toward the energy transition to more sustainable solutions for the industrial and residential heating sector [1]. According to the European Green Deal, the European Union will provide several incentives for the use of heat pumps, and it predicts that the heat pump market will show a significant increase [2]. In the manufacturing of heat pumps and electric or gas water heaters, copper coils play a crucial role. This structure is responsible for allowing water circulation and heat transfer between fluids. For this reason, it is essential that this component is long and has high thermal conductivity. Therefore, the coil is made up of a series of welded and bent copper pipes. For its use in heat pumps, it is crucial that the welding of the copper pipes can resist the high pressures and vibration inherent in these systems [3,4]. Some heat pump systems work under vacuum conditions, and leakages through the copper pipe connections will cause these systems to break down [5,6].

Copper is a metal with ductile characteristics, which makes it easily workable, although welding this material is generally challenging due to the high thermal conductivity of copper and its alloys [7,8]. Due to this characteristic condition of this type of material, it is almost always necessary to preheat the material in preparation for welding, even for low thicknesses. However, the great versatility of copper combined with its excellent electrical conductivity and ductility means that this metal is used in numerous applications, from thin sheets of copper used in small circuits of electronic components to thick plates [9,10]

used for heat dissipation of some components. Nowadays, the implementation of electrical components and devices has grown exponentially in several sectors, as is the case, for example, in the automotive sector. This increase in the demand and use of copper alloys means there is a need to improve the welding quality, efficiency, and automation of welding these alloys, for better and easier integration into an industrial process [11,12].

The welding of copper pipes is usually performed through the brazing process in a continuous furnace. In this process, the components are subjected to a long period of approximately one hour inside the 5 m long oven, where temperatures vary from approximately 710 °C to 830 °C. To guarantee the integrity of the internal environment, the oven inlets and outlets are protected by nitrogen curtains, thus preventing contamination of the gas inside. Inside the furnace, a mixture of 5% hydrogen is normally used as a shielding gas to reduce the formation of slag. After leaving the furnace, the copper coils cool naturally to room temperature. The presence of defects is usually detected only when the welded parts are subjected to a leak test closer to the end of the manufacturing line. When faulty parts appear, they must be replaced, and more resources will be wasted. Thus, it is the aim of this work to investigate the welding parameters for copper pipe brazing in order to automate and optimize the welding process, reducing the time and resources wasted caused by badly welded pipes. Extraction of welding defect information can be obtained by digital microscopy. These approaches can quantitatively relate the size and the shape defects formed during the welding process to the welding parameters and properties of the materials. This work has been a collaboration with Bosch Thermotechnology, which is one of the strongest manufacturers of heat pumps in the market. The objective is to help the industry improve its copper pipe welding processes and prevent product failures, contributing to sustainable manufacturing, cleaner production, and energy savings. This work establishes a process for improving and optimizing the system and parameters for the brazing of copper pipes in heat pump applications.

Welding is a joining process in which two or more components are joined, producing continuity in the nature of the part's materials through heat, pressure, or both. This process can occur with or without the use of filler material, with continuity in the nature of the material of the parts. Welding copper alloys can be carried out using various methods, such as laser, electron beam, resistance, friction, ultrasonic, electromagnetic pulse, and brazing [13–16]. Brazing is a process of joining materials that occur through the melting and solidification of a filler metal that is placed in contact with the base material, that is, with the components to be welded, resulting in a watertight and tight joint in a structural connection between the pieces. Brazing is usually performed with localized flame heating, oven, and induction. This process is widely used in the industry due to its great versatility [17]: namely, it allows the joining of most metals and ceramic materials; allows the joining of heterogeneous materials; it is a process that can be carried out both manually and automatically, being easily adapted to both the production of large quantities of parts and the production of individual parts; and distortions are almost zero, due to low residual stresses. However, this process also has some disadvantages, such as being able to cause corrosion; the joints must be small; and the need for careful preparation of the joint to obtain satisfactory results. The joint is highly dependent on the joint clearance or distance between parts and the optimal distance depends on the filler material and joint design [18]. To guarantee a good brazed joint, both components to be connected must be properly cleaned and protected from oxidation, either through the use of flux or through a controlled atmosphere [19,20].

Brazing flux is a mixture of chemicals used to facilitate the creation of a solid joint during the brazing process by protecting the base material and filler metal from oxidation and the formation of other undesirable substances. By removing the oxides present, the flow reduces surface tensions and promotes the free flow of the filler metal [17].

In reference [21], the influence of pressure and metal filler amount on the microstructure and strength of copper joints was studied. With this experimental activity, it was concluded that both the pressure exerted, and the amount of filler material used, will influ-

ence the characteristics of the joint. Both with low pressure and with little filler material, cavities are created in the joint, thus reducing its strength. In these cavities, when they are under stress, small cracks begin to occur, which spread and eventually cause fractures. There are studies that try to predict through finite element methods the strength of brazed joints [22].

To carry out good brazing it is imperative to have the appropriate materials, namely, filler metal compatible with the base material, but also the preparation and cleaning of the joint and protection during welding, which can be carried out in a controlled atmosphere with gas, in a vacuum, or by adding a suitable agent to the filler metal [23,24].

Another factor that influences brazing is the roughness of the joint. Several studies have been carried out to investigate the influence of roughness on the quality of welded connections; however, this is a non-consensual topic. According to reference [25], and this being the most defended hypothesis, for copper, the reduced roughness leads to a reduction in the volume of voids and an increase in the yield stress. Therefore, the smoother the base material, the greater the wetting of the joint, resulting in a higher-quality connection. But there are some articles that argue the opposite; according to reference [26], a rougher surface will generate a more turbulent flow of the filler material, and this will cause the wetting of the rough surface to be greater than that of the smoother surface. Furthermore, the rougher walls of the base material will create more metallurgical connection points, which leads to a union with higher mechanical strength.

Joint clearance plays a crucial role in generating capillary pressure, which is essential to allow the filler material to infiltrate the joint properly. The capillary pressure can be calculated using Young–Laplace Equation (1), where $p$ is the pressure, $\gamma$ is the surface tension of the filler material and $r$ is the radius of curvature of the meniscus (directly linked to the gap between surfaces to be brazed) [22].

$$\Delta p = \frac{2\gamma}{r} \tag{1}$$

Figure 1 shows the effect that the size of the joint gap has on the capillary pressure created. There is an inversely proportional relationship between the size of the gap and the capillary pressure. The vertical line represented in the graph refers to the theoretical clearance used in the specimens in this work.

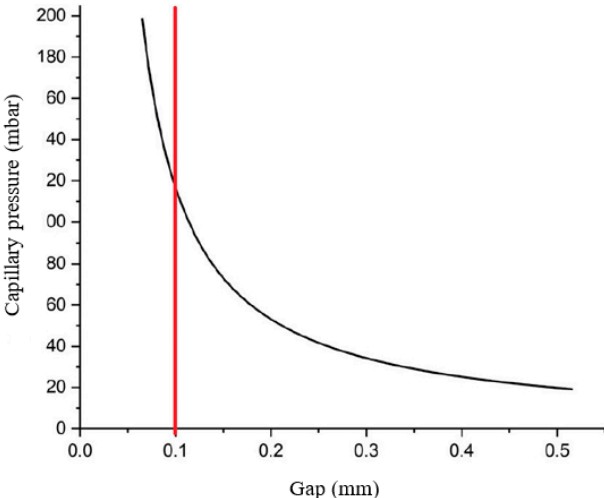

**Figure 1.** Effect of capillarity as a function of clearance (Adapted from [22]).

Regarding the length of the brazing joint, it is recommended that it be at least three times greater than the thickness of the thinnest component that will be welded [27]. This proportion ensures an adequate contact area between the materials and provides a more uniform distribution of the filler material along the joint.

## 2. Materials and Methods

### 2.1. Materials

The base material used in the tests was a copper pipe of type CW024A, rolled product with 99.90% pure copper, and the filler material is a copper–silver–phosphorous alloy, Braze CuP 281a, and its technical information can be consulted in Supplementary Material. The pipes have a nominal outer diameter of 9.55 mm and a thickness of 0.8 mm (Figure 2a). The filler metal was preformed into rings with an inner diameter of 10 mm (Figure 2b), specially designed to fit precisely on the outside of the pipe, allowing a proper connection during the brazing process. The rings were supplied with thicknesses of 1 mm and 1.5 mm, which allowed a comparative analysis of the quality of the welded joints of these two types of rings. With this material, due to its composition including phosphorus, for brazing copper alloys, it is not necessary to use any type of flux, as this chemical element already acts as a flux.

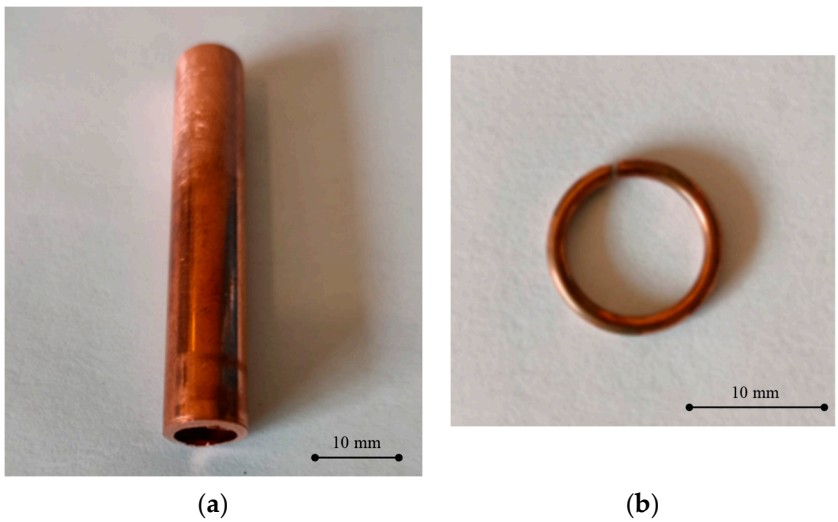

(**a**)  (**b**)

**Figure 2.** (**a**) Copper pipe (base material); (**b**) Filler material ring.

Table 1 describes the characteristic temperatures of this type of material. The term liquidus refers to the lowest temperature value under equilibrium conditions at which the metal is completely liquid and the term solidus refers to the highest temperature value under equilibrium conditions at which the metal is completely solid [17]. The technical data of the filler material is provided in Supplementary Material.

**Table 1.** Characteristics of the filler material.

| AWS Classification | Solidus (°C) | Liquidus (°C) | Brazing Temperature (°C) |
|---|---|---|---|
| Braze CuP 281a | 710 | 793 | 732–843 |

Nitrogen was used as a shielding gas, which is a low-cost inert gas widely used in the copper brazing process. Nitrogen is an inert gas for most metals; however, this gas must be used carefully when the metals to be protected have chemical elements in their composition, such as titanium or chromium. The fact that this gas is neither combustible nor explosive is a safety factor that also makes nitrogen a highly sought-after gas. Pure nitrogen is an excellent protective atmosphere for brazing copper [17]. Nitrogen plays a crucial role in preventing oxidation and contamination of materials during the welding process.

### 2.2. Tools

To enable the preparation of pipes and obtain quality and repeatable welded connections, it was essential to develop specific tools, as close as possible to those used industrially, but adapted to the size of the test pieces. The first tools developed were the die and punches, with the aim of creating gaps in the pipes. Two punches were produced, one with a diameter of ⌀9.8 mm and the other with a diameter of ⌀9.9 mm, as represented in Figure 3, to allow the creation of gaps of different sizes. This approach was able to determine which option provided the most favorable results. The development of these tools was fundamental in the pipe preparation process, aiming to achieve the objectives of quality and consistency in welded connections.

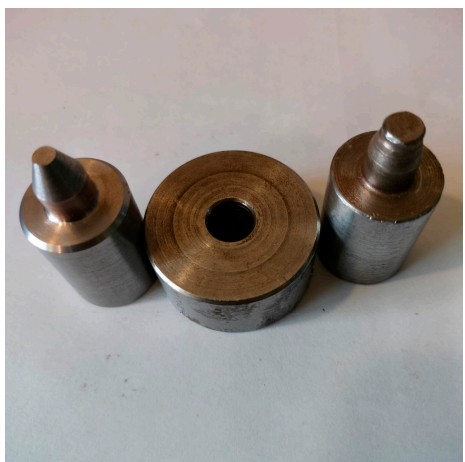

**Figure 3.** Die and punches used to create the gaps in the copper pipes.

Another necessary tool to ensure welding consistency is the jig. The jig is a device that holds the pipes during the brazing process and guarantees concentricity between the materials. Three different tools were developed for this device. Due to the ovality of the pipes, designing the jig is a complex challenge to ensure its efficient operation. The effectiveness of this tool is crucial to guarantee the quality of the welds carried out.

A tool was also developed to carry out tests on the tightness of welded connections, which consists of two plates that tighten the pipe at the ends with the help of three threaded bars and plastic sealing washers. This configuration allows the application of compressed air to test the sealing of the welds. With this tool, the pipes are isolated at the ends using rubber washers, and compressed air is injected into the pipes. The parts are then submerged in water to check the presence of air bubbles in the water. If there are no air bubbles it is guaranteed that there are no leaks in the welded pipes.

### 2.3. Material Preparation

The initial step in preparing the copper pipes consisted of cutting the pipes to the dimensions defined for the test pieces. The pipes were received as 1 m rods and were cut into small samples of 40 mm and 55 mm using a copper pipe cutter. Subsequently, the process of enlarging the pipes continued. To do this, a matrix and punches were used to form the samples with a length of 40 mm. The size of the test pieces was chosen to reduce production time and ensure a more precise fit in the matrix, minimizing possible errors. Using the Shimadzu AG-X plus 100 kN universal testing machine (Shimadzu Europa GmbH, Duisburg, Germany), the necessary compression was carried out to create the desired gap, as illustrated in Figure 4. This operation made it possible to obtain the desired dimensions in the pipes, preparing them for the subsequent stages of the brazing process.

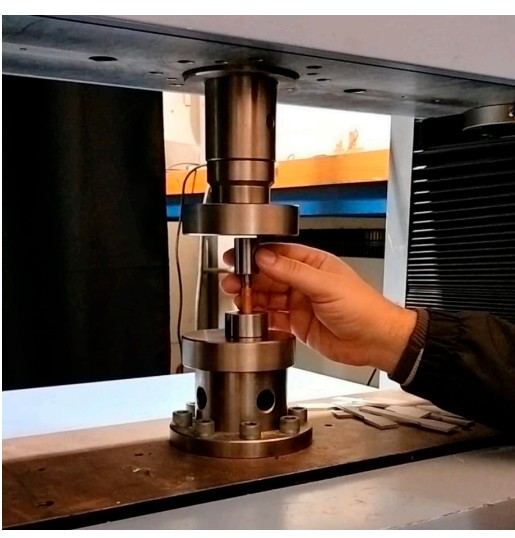

**Figure 4.** Pipe widening process.

The pipes were formed using two different punches to create gaps of different sizes in the pipes. This approach allowed for a subsequent comparative analysis to determine which measurements would be the most appropriate gap for brazing these pipes. The dimensions of the enlargements obtained are detailed in Table 2, providing precise information about the results after the pipe-forming process.

**Table 2.** Inner diameter of the pipes.

|  | Sample 1 | | Sample 2 | | Sample 3 | | Sample 4 | | Sample 5 | |
|---|---|---|---|---|---|---|---|---|---|---|
| **Position** | **0°** | **90°** | **0°** | **90°** | **0°** | **90°** | **0°** | **90°** | **0°** | **90°** |
| ∅ inner (mm) (Punch 9.8 mm) | 9.9 | 9.86 | 9.88 | 9.9 | 9.9 | 9.82 | 9.88 | 9.82 | 9.84 | 9.88 |
| ∅ inner (mm) (Punch 9.9 mm) | 10.02 | 9.9 | 9.94 | 9.96 | 9.94 | 9.88 | 9.94 | 10 | 9.98 | 9.94 |

To complete the preparation of the pipes, they were sanded in the section where the brazing was to be carried out. This step aims to ensure a suitable surface for welding, removing any impurities and promoting ideal adhesion between the materials. After sanding, the pipes were washed in running water using soap and an abrasive cloth. This procedure aims to remove any type of grease present on the surface of the pipes, thus avoiding contamination of the welding. Some rings were also cut and shaped to have an external diameter slightly less than 9 mm, as can be seen in Figure 5, so that they could be placed inside the pipes, thus brazing with an internal ring. However, it was only possible to form the 1 mm rings since the 1.5 mm rings were too rigid and broke during forming.

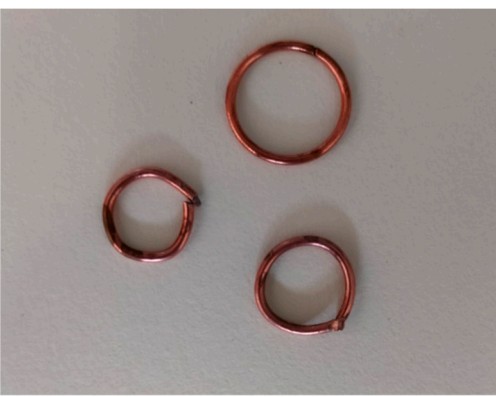

**Figure 5.** Comparison between outer and inner rings.

### 2.4. Brazing Process

The brazing process of the test pieces begins immediately after their preparation, starting with the preheating of the furnace to the established operating temperature. After the oven is at the defined temperature, the copper pipes are placed in the template together with the solder ring, which is always placed on the pipe so that when it melts it penetrates the joint, due to the capillary effect facilitated by gravity, and finally these components are placed in the oven, Figures 6 and 7.

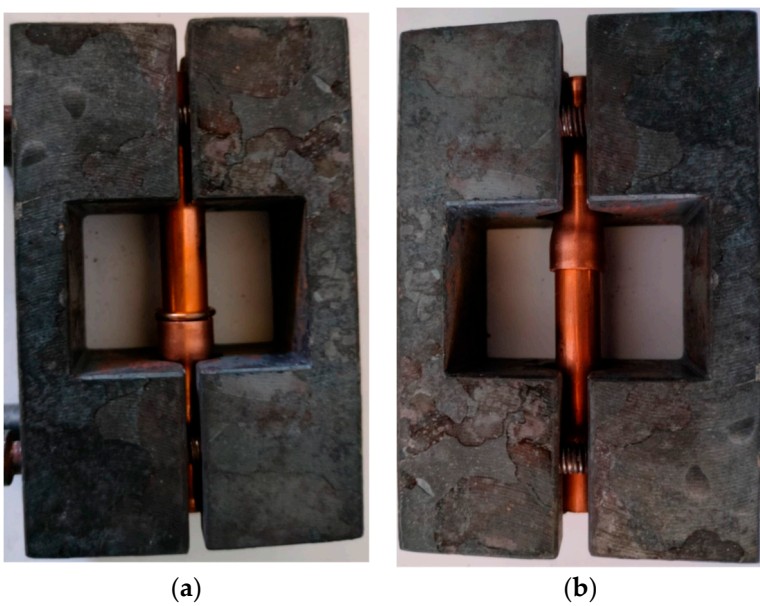

(a)          (b)

**Figure 6.** Placement of the pipe (**a**) with external ring in the jig; (**b**) with inner ring in the template.

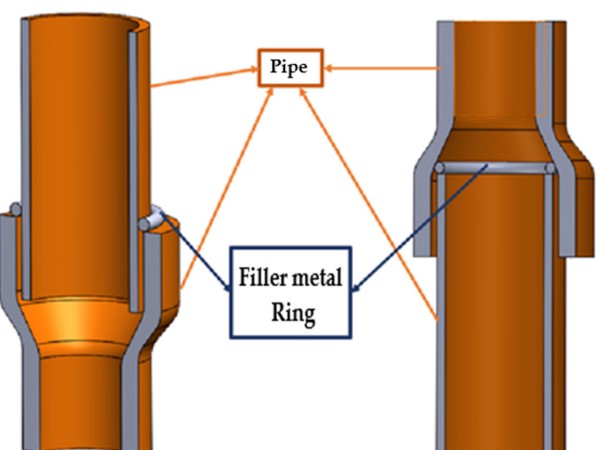

**Figure 7.** Scheme for placing the filler material.

In the first phase, the parameters used in the reference industry were replicated, with the test pieces being inserted into the oven for 50 min, with temperatures varying between 710 °C and 830 °C. As a protective gas, nitrogen was used in this experimental activity. The temperature of the furnace was measured by a thermocouple that was part of the furnace itself, which had an error of ±0.8%. In addition, another thermocouple was installed to allow a comparison with the temperature obtained with the furnace thermocouple. The difference between both thermocouples is 1 °C.

Then, to try to replicate the induction brazing process and reduce the brazing time to a minimum, an attempt to find out what was the minimum time for the filler material to melt for each brazing temperature was carried out. A glass door was placed in the oven, so

that the interior was visible, and through an iterative process, where the temperature was gradually increased, the values presented in Table 3 were reached.

**Table 3.** Brazing parameters of the test pieces.

| Sample | ⌀ Punch | Gas | Temp. °C | Time | Ring | Thickness (mm) |
|---|---|---|---|---|---|---|
| V-cg-50-1.5e-p | | Yes | 710–830 | 50 min | External | 1.5 |
| 710-sg-2.4-1.5e-p | | No | 710 | 2 min 40 s | External | 1.5 |
| 730-sg-2.3-1.5e-p | | No | 730 | 2 min 30 s | External | 1.5 |
| 770-sg-1.45-1.5e-p | | No | 770 | 1 min 45 s | External | 1.5 |
| 800-sg-1.3-1.5e-p | | No | 800 | 1 min 30 s | External | 1.5 |
| 800-sg-1.3-1i-p | 9.8 | No | 800 | 1 min 30 s | Internal | 1 |
| 810-sg-1.2-1.5e-p | | No | 810 | 1 min 20 s | External | 1.5 |
| 820-sg-1-1.5e-p | | No | 820 | 1 min | External | 1.5 |
| 820-sg-1-1e-p | | No | 820 | 1 min | External | 1 |
| 820-sg-1-1i-p | | No | 820 | 1 min | Internal | 1 |
| 860-sg-1-1.5e-p | | No | 860 | 1 min | External | 1.5 |
| 860-sg-1-1i-p | | No | 860 | 1 min | Internal | 1 |
| 820-sg-1-1.5e-g | 9.9 | No | 820 | 1 min | External | 1.5 |
| 820-sg-1-1i-g | | No | 820 | 1 min | Internal | 1 |

Three specimens were produced for each sample carried out, resulting in a total of 42 specimens. In Table 3, the name of the samples was given to indicate the parameters used, temperature—shielding gas (cg—with gas; sg—no gas)—brazing time for complete visual fusion—placement of the ring—diameter of the ring wire section.

## 2.5. Joint Quality Check Tests

To test the quality of welded connections, four methods were used in addition to visual inspection.

### 2.5.1. Tightness Test

The first test carried out on all samples was the tightness test. In this test, each sample was inserted into a tool, as shown in Figure 8, and compressed air was injected into the sample, with a pressure of 8 bar. Then, the assembly was submerged in water, and it was checked whether air bubbles were forming. If no air bubbles were detected, the sample was considered watertight.

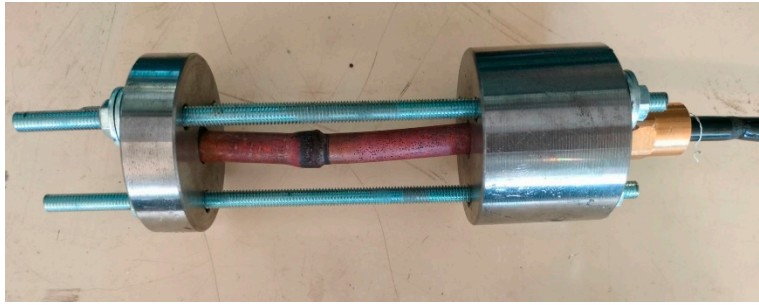

**Figure 8.** Tool for tightness testing of the samples.

### 2.5.2. Electron Microscopy (SEM and EDS)

The SEM has the advantage of high spatial resolution and the possibility of allying semi-quantitative chemical analysis (EDS) techniques to determine the chemical composition of the sample. Samples were cut transversely and were mounted on the scanning electron microscope, a Hitachi TM4000Plus (Hitachi High-Tech Corporation, Tokyo, Japan), equipped with a Bruker Quantax 75 EDS (Bruker Corporation, Billerica, MA, USA). The

microscope was set to 1000× magnification using SEM electron detector (Hitachi High-Tech Corporation, Tokyo, Japan) at 15 mm working distance [28,29].

### 2.5.3. Optical Microscopy

The samples were cut with a metallographic saw and mounted in epoxy resin. After the required grinding (water-cooled manual grinding with SiC abrasive papers with 320, 400, 800, 1200, and 2400 grit) the samples were observed in a stereo microscope Leica EZ4 W (Leica Microsystems, Wetzlar, Germany) with a 5-megapixel camera and 8× to 35× magnification. Lower magnification images were acquired for a general understanding of the state of the welding while images at higher magnifications were acquired for digital processing and image analysis. The defects existing in the welding connection joint were inspected in the cross sections of the tubes. The defects quantification and qualification were determined by digital image analysis using NIH Image J public domain software (version 1.54g). This routine is described in detail elsewhere [30].

### 3. Results and Analysis

#### 3.1. Tightness Test

The tightness test was carried out at a pressure of 8 bar on all specimens, before they were subjected to any other type of test. The results obtained in this test are shown in Table 4, where 92% of the brazed pipes are watertight. The term OK in Table 4 means that the sample did not leak air when the pressure inside the pipe was 8 bar. Of the samples that did not have positive results in terms of tightness, two of them did not completely melt the filler material, and in one of the samples, part of the filler ring material melted out of the joint.

**Table 4.** Tightness test results.

| Sample | Watertight | | |
|---|---|---|---|
| V-cg-50-1.5e-p | OK | OK | OK |
| 710-sg-2.4-1.5e-p | OK | NOK | OK |
| 730-sg-2.3-1.5e-p | OK | OK | NOK |
| 770-sg-1.45-1.5e-p | OK | OK | OK |
| 800-sg-1.3-1.5e-p | OK | OK | OK |
| 800-sg-1.3-1i-p | OK | OK | OK |
| 810-sg-1.2-1.5e-p | OK | OK | OK |
| 820-sg-1-1.5e-p | OK | OK | OK |
| 820-sg-1-1e-p | NOK | OK | OK |
| 820-sg-1-1i-p | OK | OK | OK |
| 860-sg-1-1.5e-p | OK | OK | OK |
| 860-sg-1-1i-p | OK | OK | OK |
| 820-sg-1-1.5e-g | OK | OK | OK |
| 820-sg-1-1i-g | OK | OK | OK |

#### 3.2. Quantitative and Qualitative Analysis of Welding Quality

Figure 9 exhibits an example sample sectioned longitudinally. One can observe that in this operation the penetration of the addition material is continuous throughout the entire piece.

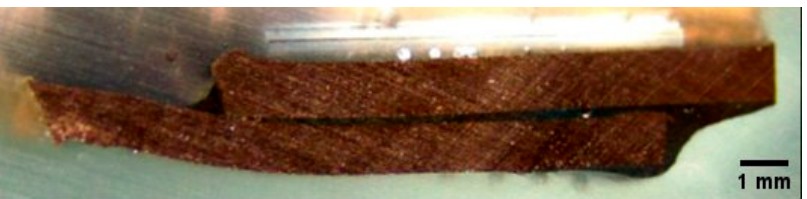

**Figure 9.** Longitudinal section of the joint.

The global results of the quantitative analysis of connection joints can be seen in Figures 10–12 and Tables 5 and 6. In Figure 10, it is possible to visualize a first analysis of factors such as the uniformity of the filler material along the joint and the homogeneity of the gap size. Figure 11 shows small defects in the filler materials. Figure 12 shows the different joint gap thicknesses from the calculated area.

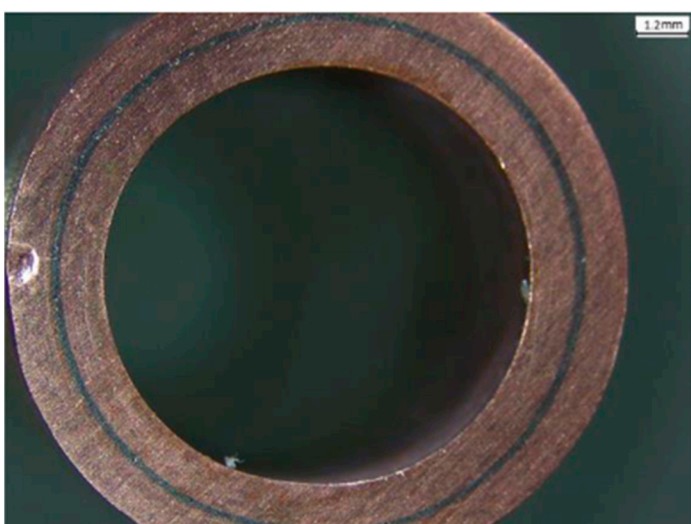

**Figure 10.** Stereo microscope image of the pipe cut transversely to the brazed connection.

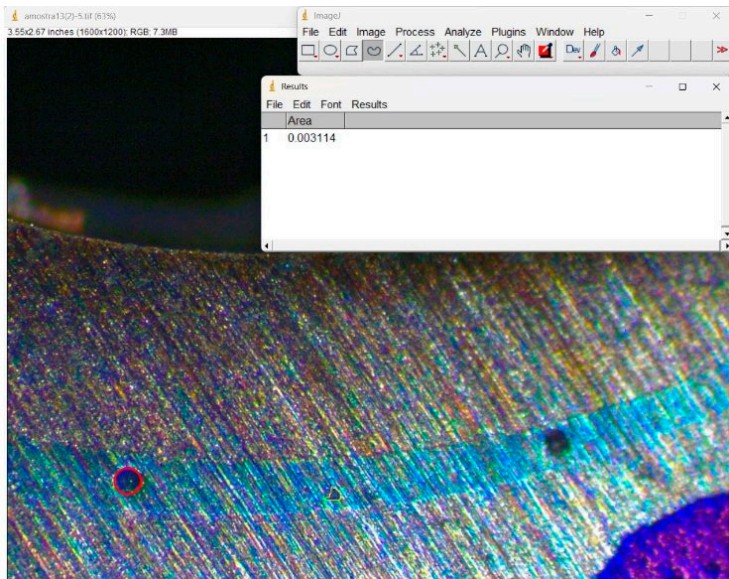

**Figure 11.** Examples of small defects circled in red, and their respective area values, calculated from the image J program.

The results presented in Table 5 correspond to the combination of information obtained from the cross sections of the samples, where $\varepsilon_{max}$ and $\varepsilon_{min}$ respectively are the maximum and minimum values of welding joint clearance; $\varepsilon_{gap}$ is the thickness of the clearance in which these defects occur; and Area is the total area of material failures. The data obtained are presented in Table 5. A lack of material is understood to be a joint defect, in which there is no filler material between the two welded components in a given section of the sample, such as the defects highlighted in Figure 13. The part presented in Figure 13 has failed the leakage test due to the lack of filler material.

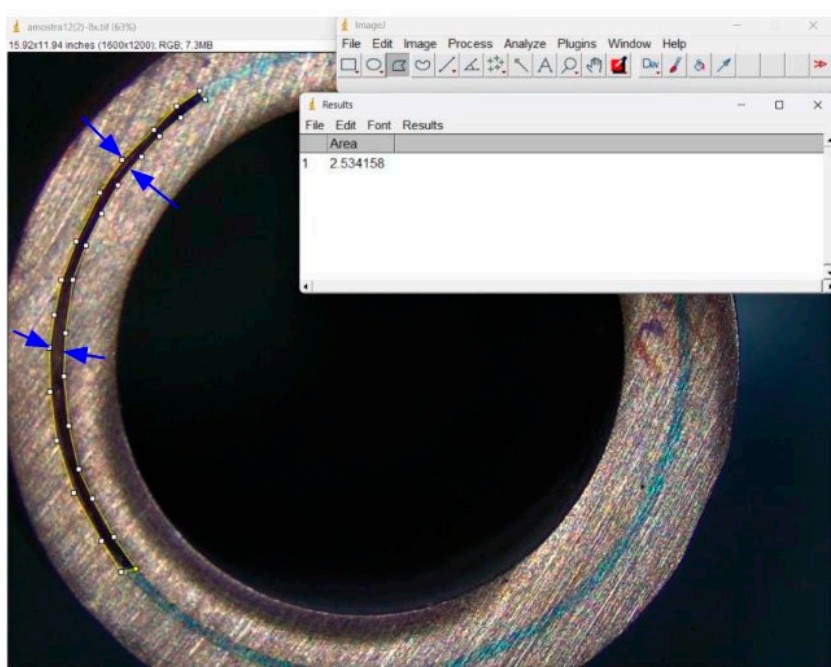

**Figure 12.** Example of the existence of difference gap thickness that causes heterogeneities in the thickness of the welded joint.

**Table 5.** Actual clearance size of the test pieces.

| Sample | $\varepsilon_{max}$ (mm) | $\varepsilon_{min}$ (mm) | Lack of Material | $\varepsilon_{gap}$ (mm) | Area (mm$^2$) |
|---|---|---|---|---|---|
| V-cg-50-1.5e-p | 0.42 | 0.08 | Yes | 0.1 | 0.119 |
| 710-sg-2.4-1.5e-p | 0.3 | 0.12 | Yes | 0.3 | 1.175 |
| 710-sg-2.4-1.5e-p | 0.33 | 0.12 | Yes | 0.3 | 1.213 |
| 730-sg-2.3-1.5e-p | 0.25 | 0.1 | Yes | 0.1 | 0.5 |
| 730-sg-2.3-1.5e-p | 0.27 | 0.13 | Yes | 0.27 | 0.76 |
| 770-sg-1.45-1.5e-p | 0.22 | 0.07 | Yes | 0.22 | 0.41 |
| 770-sg-1.45-1.5e-p | 0.2 | 0.06 | No | - | 0 |
| 800-sg-1.3-1.5e-p | 0.29 | 0.07 | Yes | 0.29 | 3.77 |
| 800-sg-1.3-1.5e-p | 0.17 | 0.08 | Yes | 0.16 | 0.68 |
| 800-sg-1.3-1i-p | 0.27 | 0.24 | Yes | 0.27 | 1.97 |
| 800-sg-1.3-1i-p | 0.29 | 0.12 | Yes | 0.29 | 2.53 |
| 810-sg-1.2-1.5e-p | 0.11 | 0.07 | Yes | 0.09 | 1.08 |
| 810-sg-1.2-1.5e-p | 0.27 | 0.06 | Yes | 0.27 | 1.55 |
| 820-sg-1-1.5e-p | 0.26 | 0.06 | Yes | 0.26 | 1.6 |
| 820-sg-1-1.5e-p | 0.37 | 0.08 | Yes | 0.37 | 2.44 |
| 820-sg-1-1e-p | 0.24 | 0.18 | No | - | 0 |
| 820-sg-1-1e-p | 0.26 | 0.14 | No | - | 0 |
| 820-sg-1-1i-p | 0.3 | 0.09 | Yes | 0.3 | 2.46 |
| 820-sg-1-1i-p | 0.25 | 0.06 | No | - | 0 |
| 860-sg-1-1.5e-p | 0.3 | 0.1 | Yes | 0.3 | 1.86 |
| 860-sg-1-1.5e-p | 0.34 | 0.08 | Yes | 0.34 | 2.8 |
| 860-sg-1-1i-p | 0.27 | 0.14 | Yes | 0.27 | 0. |
| 860-sg-1-1i-p | 0.25 | 0.06 | No | - | 0 |
| 820-sg-1-1.5e-g | 0.61 | 0.03 | Yes | 0.61 | 2.11 |
| 820-sg-1-1.5e-g | 0.65 | 0.02 | Yes | 0.65 | 4.65 |
| 820-sg-1-1i-g | 0.28 | 0.24 | No | - | 0 |
| 820-sg-1-1i-g | 0.3 | 0.23 | No | - | 0 |

**Table 6.** Defects per test piece.

| Sample | Large Defects | Small Defects |
|---|---|---|
| V-cg-50-1.5e-p | 27 | 14 |
| V-cg-50-1.5e-p | - | - |
| 710-sg-2.4-1.5e-p | 54 | 16 |
| 710-sg-2.4-1.5e-p | 47 | 21 |
| 730-sg-2.3-1.5e-p | 68 | 7 |
| 730-sg-2.3-1.5e-p | - | - |
| 770-sg-1.45-1.5e-p | 40 | 14 |
| 770-sg-1.45-1.5e-p | 32 | 22 |
| 800-sg-1.3-1.5e-p | 38 | 16 |
| 800-sg-1.3-1.5e-p | 28 | 10 |
| 800-sg-1.3-1i-p | 44 | 10 |
| 800-sg-1.3-1i-p | 49 | 8 |
| 810-sg-1.2-1.5e-p | 42 | 12 |
| 810-sg-1.2-1.5e-p | 35 | 9 |
| 820-sg-1-1.5e-p | 27 | 10 |
| 820-sg-1-1.5e-p | 25 | 19 |
| 820-sg-1-1e-p | 23 | 10 |
| 820-sg-1-1e-p | - | - |
| 820-sg-1-1i-p | 44 | 16 |
| 820-sg-1-1i-p | 65 | 18 |
| 860-sg-1-1.5e-p | 33 | 14 |
| 860-sg-1-1.5e-p | - | - |
| 860-sg-1-1i-p | 37 | 21 |
| 860-sg-1-1i-p | 33 | 16 |
| 820-sg-1-1.5e-g | 31 | 6 |
| 820-sg-1-1.5e-g | 37 | 9 |
| 820-sg-1-1i-g | 28 | 7 |
| 820-sg-1-1i-g | 37 | 15 |

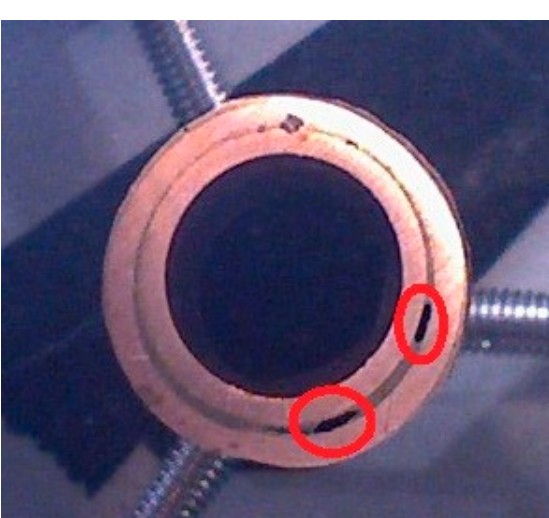

**Figure 13.** Example of a part that has failed the leakage test due to a lack of filler material.

In this analysis, defects were divided into two categories. Larger defects are malformations that have a value equal to or greater than 0.003 mm$^2$. Small defects are malformations measuring less than 0.003 mm$^2$. This decision to classify the defects as small or large was made based on an optical microscope magnification threshold of 20×. The accounting of all defects is presented in Table 6.

### 3.3. Scanning Electron Microscopy (SEM) and Energy Dispersive X-ray Spectroscopy (EDS)

Electron microscopy and microanalysis were performed to complement the qualitative analysis obtained by optical microscopy. Two specimens were analyzed using Scanning Electron Microscopy, a specimen from sample 1, the reference or control sample, and a specimen from sample 8. It was possible to detect three types of defects existing in the addition material [31].

The first type of defect is empty spaces in the addition material. These empty spaces have different dimensions, ranging from pockets with diameters of a few micrometers as in Figure 14a, to large flaws that are visible to the naked eye, Figure 13. The portion marked in these figures is empty space. These empty spaces could have been caused by different factors, e.g., gas trapped inside the joint may have caused the smaller flaws; however, the large flaws visible to the naked eye were caused by the movement of the components with the filler material still in its fluid state, in the process of removing the specimens from the furnace.

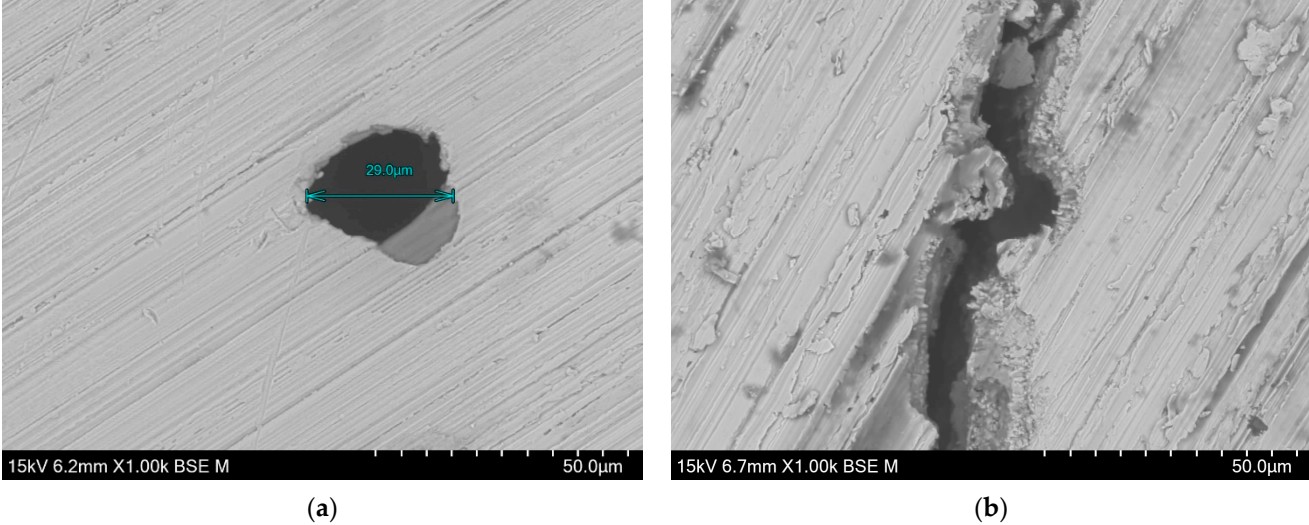

(**a**)        (**b**)

**Figure 14.** SEM image of sample 8 with 1000× magnification of (**a**) small empty space in the addition material; (**b**) crack in the connection joint.

The second type of defect that was also identified through the SEM analysis is cracks, which is represented in Figure 14b. These cracks occur at the border between the base material and the addition material, and due to their contour, they can also be created with the movement generated when removing the sample from the furnace. In addition, the combined effect of the initial deformation caused by the placement of the pipes in the template and the thermal stress resulting from the heating and cooling process when removing the joint from the furnace can also create these cracks.

In addition, an EDS test was carried out on a specimen of sample 13, which allowed the identification of the presence of chemical elements as the third type of defect. The section of the specimen used in this process is represented in Figure 15. Due to the visual characteristics, it is possible to distinguish the base material (copper) from the filler material. The border between the two regions is marked by the red line in Figure 15a. By analyzing the EDS image from Figure 15, it is possible to identify some regions where defects are present in the addition material. The chemical composition of these regions was analyzed aiming to comprehend the cause of the defects.

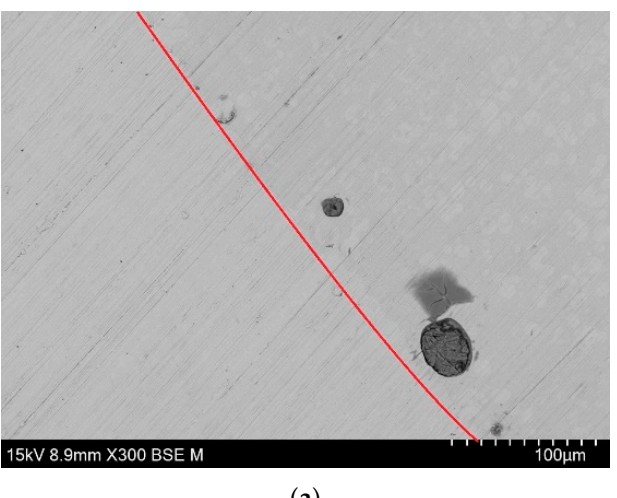
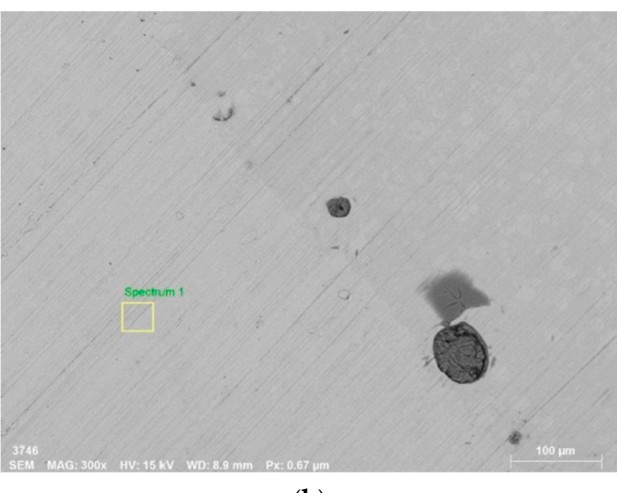

(**a**)

(**b**)

**Figure 15.** Image of sample 13 with $300\times$ magnification: (**a**) Border between the two regions marked by the red line; (**b**) Region in the base material where the EDS process was carried out, marked by the yellow square.

The first analysis carried out was on the base material side, in the region marked by the yellow square in Figure 15b. The results obtained are shown in Table 7. The sample is composed of approximately 91% of copper, 8% of carbon, and 1% of oxygen.

**Table 7.** Chemical components on the base material side.

| Element | Atomic Number | Mass in Sample (%) |
| --- | --- | --- |
| Copper (Cu) | 29 | 91.22 |
| Carbon (C) | 6 | 8.06 |
| Oxygen (O) | 8 | 0.81 |

The chemical composition in the addition material region was also analyzed. The analyzed region is marked by the yellow square in Figure 16a. The results are presented in Table 8. It is observed that it is also mainly composed of copper; however, in addition to oxygen and carbon, a significant percentage of phosphorus is also found, which was expected as it agrees with the technical information provided by the manufacturer of the addition material.

**Table 8.** Chemical components of the filler material region.

| Element | Atomic Number | Mass in Sample (%) |
| --- | --- | --- |
| Copper (Cu) | 29 | 84.32 |
| Carbon (C) | 6 | 8.42 |
| Oxygen (O) | 8 | 1.01 |
| Phosphorus (P) | 15 | 6.26 |

Finally, the chemical composition of a defect in the joint, marked by the yellow square in Figure 16b, was analyzed. The results are reported in Table 9. In addition to the copper, carbon, and phosphorus present in the other regions, the defect region presents very high values of oxygen, around 26%, which indicates that this defect was caused by oxidation and that an oxide was formed in that pocket of material.

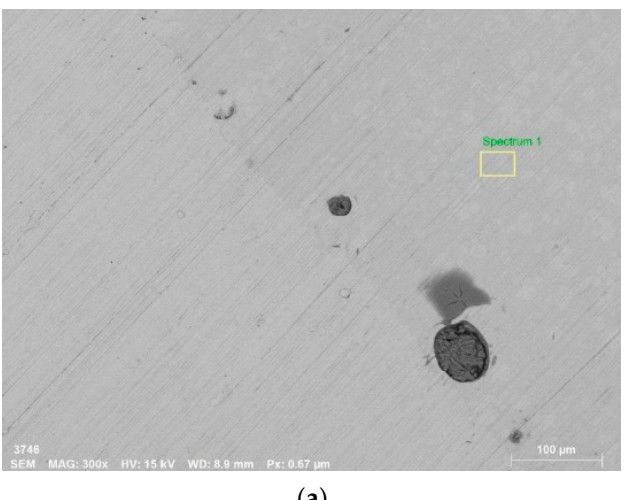
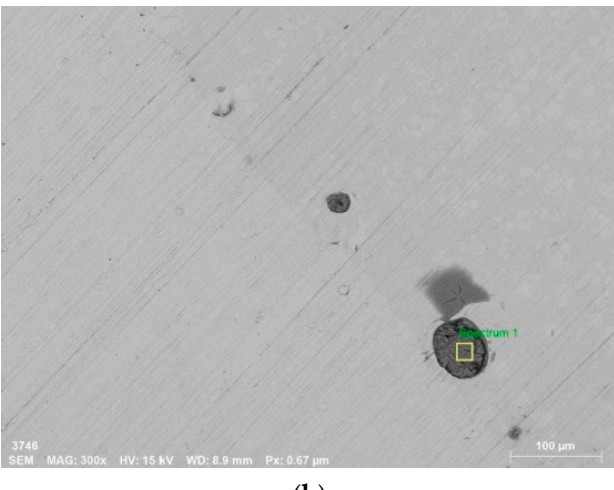

(**a**)                                                                                      (**b**)

**Figure 16.** Image of sample 13 with $300\times$ magnification: (**a**) Region where the chemical composition of the addition material was analyzed, marked by the yellow square; (**b**) Defect area considered in the chemical analysis, marked by the yellow square.

**Table 9.** Chemical components of the defect region.

| Element | Atomic Number | Mass in Sample (%) |
|---|---|---|
| Copper (Cu) | 29 | 44.67 |
| Carbon (C) | 6 | 11.17 |
| Oxygen (O) | 8 | 26.16 |
| Phosphorus (P) | 15 | 15.10 |

## 4. Discussion

### 4.1. Effect of Gap

From the data obtained in this analysis, it is possible to verify that the size of the gap is related to the lack of material in the joint. By dividing the number of gap defects existing in different clearance dimensions by the total number of defects of this type, in Figure 17, it is seen that around 17.6% of defects occur when the clearance dimension is equal to or less than 0.1 mm and 73.7% of defects occur when the gap between components is greater than 0.25 mm. Of these 73.7%, 50% of defects occur when the gap is equal to or greater than 0.3 mm. For values between 0.39 mm and 0.59 mm, there are no values recorded. These results are in agreement with other studies in the literature that mention the gap between the parts to be welded has a significant influence on the quality of the welding [17].

In Figure 18, it can be seen that the type of filler material ring also has an influence on the appearance of defects due to lack of material. For a brazing temperature of 820 °C, the thinner external filler material ring leads to greater defects due to lack of filler material. However, the application of a thicker external filler material ring causes these types of defects to have a very substantial reduction in size, and in some specimens, they do not even exist. This may be due to the fact that a thicker filler ring has more material and will better fill the gap between the pipes.

Figure 19 shows that there is a variation in the number of defects considering the size of the gap. For external filler material rings, an increase in the number of small defects can be seen when the joint gap is greater. However, using a larger gap the number of large defects is significantly lower. For filler rings placed internally in the joint, it is possible to observe a reduction in both the number of small and large defects for joints with a larger gap. The appearance of less large defects when a larger gap is used can be given by the fact that the filler material will more easily penetrate a larger gap than a small gap, which will lead to a reduction in filler material voids in the welding joint.

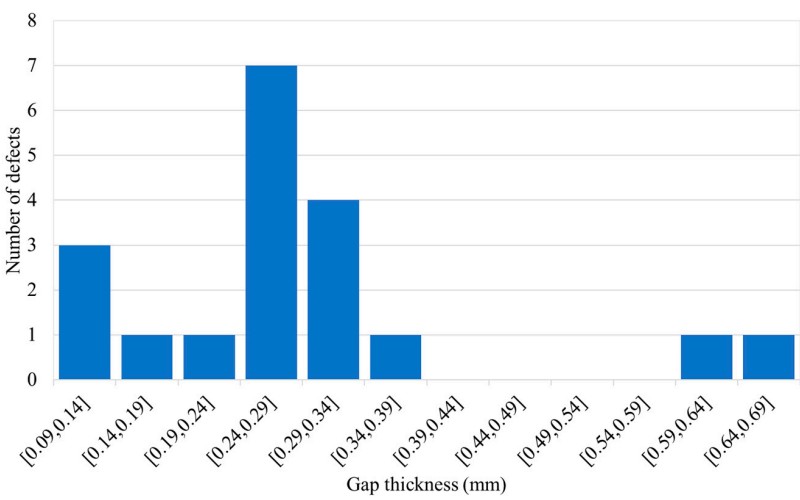

**Figure 17.** Number of defects due to lack of material by gap size.

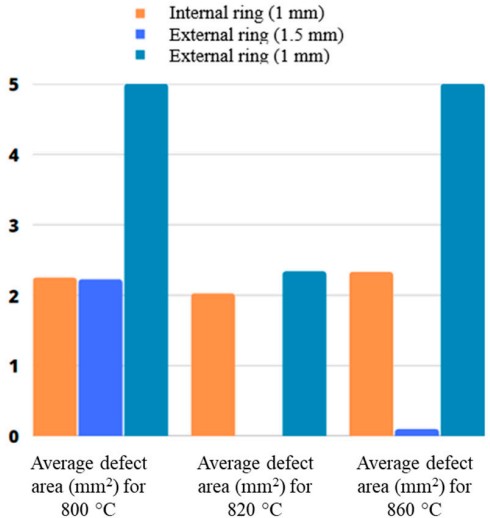

**Figure 18.** Average defect area (mm$^2$) of missing material by type of filler ring.

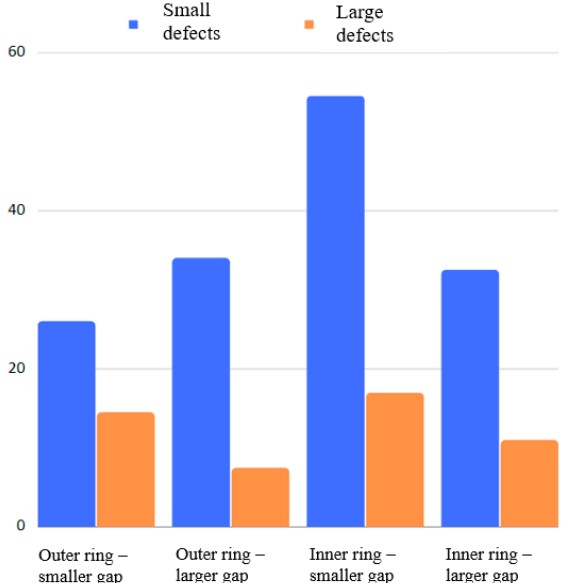

**Figure 19.** Number of defects per gap created.

## 4.2. Effect of Temperature

Figure 18 shows the average number of defects for each sample, depending on the temperature at which they were welded. For the comparison to be coherent, in this analysis, only samples enlarged with a ⌀9.8 mm punch and which were brazed with a 1.5 mm ring of external filler material were used.

From Figure 20 it is possible to see that the number of small defects in the sample decreases with temperature up to a maximum temperature of 820 °C, a temperature very similar to the 830 °C announced by the filler material manufacturer as the maximum brazing temperature. Most small defects are pockets of a lack of addition material. The cause for several of these defects may be related to the capillarity effect, which prevents the liquid addition material from properly flowing to fill the gap between the pipes. As the temperature increases, the surface tension decreases, and the liquid addition material has a lower ability to resist the pull of gravity and will flow more easily into the gap between the pipes. Large defects, on the other hand, have a less regular pattern, and the number of these types of defects does not follow a trend according to the variation in temperature and brazing time.

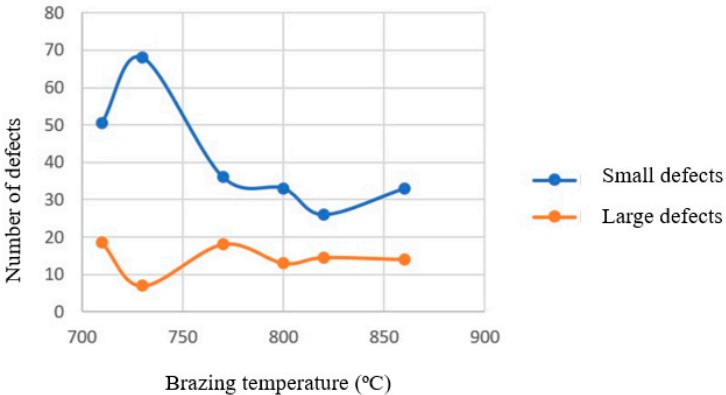

**Figure 20.** Number of defects depending on brazing temperature.

## 4.3. Effect of Addition Ring Type

Another important factor to analyze is the influence that the quantity and way of placing the filler material have on the quantity and size of the defects. In Figure 21, the average number of defects present in the specimens of samples 8, 9, and 10, respectively, is represented. These samples were chosen for this analysis due to their uniformity in terms of brazing parameters, such as brazing time and temperature, one minute at 820 °C, to allow a comparison without other types of variables. By analyzing these data for gaps of this size with the use of external filler material rings, it can be seen that the number of defects is much lower than the number of defects existing in the specimens that were brazed with internal filler material rings. Another interesting fact that was found is that there is a small decrease in the number of defects in samples in which 1 mm thick rings were used compared to samples in which the rings were 1.5 mm. However, as can be seen in Figure 18, despite the small reduction in the number of defects when using a thinner external filler ring, the average defect area is much smaller when a thicker external filler ring is used.

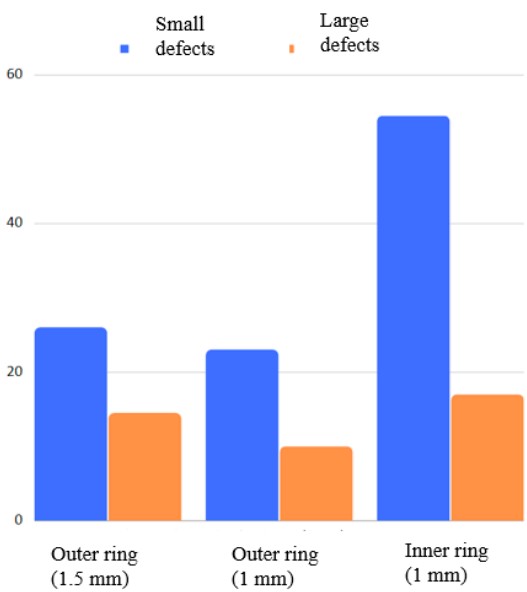

**Figure 21.** Number of defects per type of filler material ring (brazing temperature, 820 °C).

## 5. Conclusions

This work presents an analysis of several parameters for the brazing of copper pipes for heat pump applications. The welded samples were analyzed by optical microscopy, SEM, and EDS, allowing the identification of defects formed during the welding process. The effect of the parameters on the appearance and type of defects was carried out.

Optical microscopy made it possible to understand the general condition of the circular and longitudinal sections of the weld. On the other hand, the use of SEM and EDS techniques, in addition to enabling greater magnification than optical microscopy, allows an analysis of the chemical composition of the sample, which provides an understanding of the composition of the defect, allowing its categorization.

With the results obtained, it was clear that the use of a template that guarantees the concentricity of the pipes is important to obtain good results; however, when the filler material rings are placed internally, the gap size is much more uniform, which means that the number of large defects is smaller than in specimens with external rings.

The size of the gaps in the joints is a very important factor in obtaining good results, since large defects most often occur in sections where the gap is greater than 0.25 mm. Therefore, the use of a template that guarantees the concentricity of the pipes is also an important factor in obtaining satisfactory results. The variation in brazing temperature does not seem to have a major impact on the creation of large defects; however, regarding the number of smaller defects, there is a reduction with increasing temperature, up to the maximum brazing temperature of the filler material.

Defects caused by oxidation can be easily eliminated by using a protective atmosphere. However, the creation of a protective atmosphere leads to a very significant increase in production costs, which should be further studied and in-depth.

Analyzing the obtained results, it is found that using an external filler material ring of 1.5 mm diameter leads to a smaller average area of defects. However, the test pieces that showed a smaller number of defects were those subjected to a temperature of 820 °C with a 1 mm thick external addition ring. Although the results presented by these types of test pieces demonstrate some defects due to lack of material, this is largely due to the size of the gap, and it is these test pieces that present the least number of defects.

**Supplementary Materials:** The following supporting information can be downloaded at: https://www.mdpi.com/article/10.3390/met14020171/s1.

**Author Contributions:** Conceptualization, A.B.P., J.M.S.D., N.M.S. and S.D.; Formal analysis, A.B.P., J.M.S.D., J.P.R. and A.H.; Investigation, A.B.P., J.M.S.D., J.P.R., N.M.S. and S.D.; Methodology, A.B.P., J.M.S.D., J.P.R., N.M.S. and S.D.; Resources, A.B.P. and A.H.; Supervision, A.B.P. and A.H.; Validation, A.B.P., N.M.S. and S.D.; Writing—original draft, A.B.P.; Writing—review and editing, A.B.P., J.M.S.D. and A.H. All authors have read and agreed to the published version of the manuscript.

**Funding:** This research was funded by PRR—Plano de Recuperação e Resiliência under the Next Generation EU from the European Union, Project "Agenda ILLIANCE" [C644919832-00000035 | Project n° 46] and supported by the Centre for Mechanical Technology and Automation (TEMA) through the projects UIDB/00481/2020 and UIDP/00481/2020.

**Data Availability Statement:** The original contributions presented in the study are included in the article, further inquiries can be directed to the corresponding author.

**Acknowledgments:** The present study was developed in the scope of the Project "Agenda ILLIANCE" [C644919832-00000035 | Project n° 46], financed by PRR—Plano de Recuperação e Resiliência under the Next Generation EU from the European Union, and had laboratory support from the Centre for Mechanical Technology and Automation (TEMA), projects UIDB/00481/2020 and UIDP/00481/2020.

**Conflicts of Interest:** The authors declare no conflicts of interest.

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
