# Peer review of "Brazing of Copper Pipes for Heat Pump and Refrigeration Applications"

_metals, doi:10.3390/met14020171_

Round 1

Reviewer 1 Report

Comments and Suggestions for Authors

Line no 11 in abstract what is the system author refer in the study? Line no 18-19 phrases are need to be change. The interval of temperature used in the studies for brazing need to be highlighted in abstract. If possible, kindly include the details of salient findings in abstracts.

I would suggest to the authors that the introduction must be a single component rather than split it to the sections such as 1.1 and 1.2. it will be quite difficult for the readers to specify the things.

Avoid the words we in abstract and when ever and where ever a component is placed, kindly place a scale to capture the picture. Remove the fig 2 and incorporate the changes suggested. A phase diagram must be refered and placed to appreciate the liquidus temperature in the study.

section 2.2 can be cut shorted. In what aspects the fig 4 and 5 related to this work. improve the figure and table captions in the work. Suggestions is to keep the fig 7 to 9 as a single image. A normal orthogonal test rig can be prepared in the place of table 3 to optmizie the test conditions.  The authors must sit together and align the figure. the vast experimental details and the too much no of figure can be club and cut shorten.

Fig 13-16 can be ordered as a single fig. Fig 18-22 must be plotted well or other software to represent the data. The tables must be aligned properly. Similarly, all SEM must be place it together for comparison.

Over all the results were just reported and not discussed well and the science connectivity is missing in the article.

Comments on the Quality of English Language

Use MDPI language service to improve the manuscript

Author Response

The authors kindly acknowledge the Reviewers’ valuable comments and suggestions, which contributed to clarify and improve the paper.

Concerning the Reviewers’ specific comments/suggestions, the comments and actions made concerning each of them are detailed in the .word file attached.

Reviewer 2 Report

Comments and Suggestions for Authors

- Section 2.5.1

I suggest to report the standard or a reference for this test in terms of both scheme and results' presentation. 

- Section 3.1

Is it possible to see the photos of the samples that have negative result by evidencing the reason of failure?

- Section 3.2

If the computed tomography does not show detailed results, its presence in the work should be useless. I suggest or to eliminate it or to give it a different role in the whole discussion.

- Section 3.3

This part seems the more important of the work because it evidences the goodness of the joint by showing the presence of defects. I suggest to improve it also with other images by better correlating with sections 4.1, 4.2 and 4.3.

- Section 4.2

The title is effect of temperature? 

Why for low temperatue the presence of small or large defect change? What is the effect of temperature or small or large defects? I suggest to better describe this also with the aid of literature. 

- Section 4.4

Why this section is not reported in 3. Results and analysis?

- 4. Discussion

I suggest to improve this section by engaging with literature so to validate the sentences of the Authors. 

Author Response

(The authors gave the same response as above.)

Reviewer 3 Report

Comments and Suggestions for Authors

In this study, an analysis was carried out to determine which welding methods are most suitable for copper. Optimal joint shapes, process time reduction, brazing of copper alloy samples, leakage tests and micrographs and microcomputed tomography of welding samples were also investigated. The topic is interesting and has practical implications.For or a more comprehensive understanding of the problem, in my opinion, it is worth expanding the following threads:

1. The abstract of the paper needs to be modified to include the final results obtained by the authors of the paper. 2. The introduction lacks a justification of the purpose of the work. The text preceding the objective contains only an abbreviated description of the copper tube brazing process. A specific analysis of the currently used brazing methods with a clear identification of the advantages and disadvantages of these techniques is missing. Also, the area in which the authors foresee the possibility of improving the soldering technique is not specified. 3. The keywords are worth supplementing with an indication of the copper alloy used for the tubes. The use of the key word 'heat pumps' is over the top. The content of the article, apart from the introduction, has nothing to do with this term 4. As can be seen from the text figure one is borrowed from [8], so it is worth duplicating references to [8] in the caption. 5.Page 4, line 128... "sheet B...". This sheet is missing from the text of the article 6. An important part of preparing the tubes for brazing is to make the slots, keeping the tubes concentric and minimizing the ovality effect. Using a loose punch and die requires perfect alignment of all components. Using a universal punch and die to produce pressure makes dimensional repeatability very difficult to achieve. It is much easier to achieve repeatability when using a device design that provides for rigid coupling of the punch and die with the ability to move the punch along guides. As a result, the question arises whether the authors have specified a minimum maximum deviation of tube dimensions from circularity? 7 Figure 4 of the tube alignment jig shows 4 holes with a diameter of 6 mm. On the other hand, figure 8 which shows the device with the tubes assembled shows the connection with screws. Are the threaded holes made in one half of the structure? With what accuracy of concentrically were the holes made, and how was the clamping of the individual screws controlled to avoid unwanted deformation during assembly and heating? 8. How and with what accuracy was the temperature in the furnace controlled? 9. In figure 19 the required editorial right of area units also in the caption the 2 must be given as the upper symbol. 10. The crack defect in figure 24 is rather due to the combined effect of the initial deformation caused by the placement of the tubes in the template and the thermal stress resulting from the heating and cooling process when removing the joint from the furnace. Water leakage testing with insufficient cooling of the joint can also be a good cause of this type of defect. 11 The literature list needed to be supplemented to give a broader view of the problem under consideration and, at present, ways to improve this type of joint.

Comments on the Quality of English Language

please check the text carefully, especially with regard to editing

Author Response

(The authors gave the same response as above.)

Round 2

Reviewer 1 Report

Comments and Suggestions for Authors

Dear Authors,

The comments are as follows,  

Why did the author prefer to weld the copper material through the brazing process? There are other potential  processes available in Metal joining. The author must emphasize their application and from their point of view, it is expected to discuss in detail.  

May I know what is the max temperature and volume of the open oven used in the study. On what basis did the authors choose the temperatures?    

It is expected to reframe the line no 20-22 in abstract. How did the authors control the flow of H2 accurately in the furnace.  

Line no 51-53 is not reqd in the topic of discussion with reference to the defects.  

Line no 62-82 many of them known about the welding and brazing process. Why did the authors keen interest to discuss from definition. I think some recent literature can be included rather than definition.  

What is the source of device used to do the capillary force measurement in work.  

Line no 145 why did the authors discussed about other material when they have selected copper as a base material in the work and the discussion with reference to N2 as a source here.  

Table 2 std deviation is to be followed .  

The ASTM test method should be follow for tightness test and also the authors must discuss the importance of the test in application point of view. Also, the authors must stated the OK means acceptable and in what aspects it satisfy the test condition in the study  

Fig 9 to 12 can be merge it together with single caption rather than mentioned it as a seperate figure with least discussion.  

None of the defects in the test rigs are not able to spot out through any of the characterization techniques. The SEM analysis data in the work is very poor and there is no much information of the captured data from the machine. 

The Mass% of sample in the study is not accurate in EDS. The reliability of the data leads to questionable.  

The objective of gap and the study is not clearly stated and it is difficult to identify and correlate with the existing results.  

The outline of the paper must clearly stated and the discussion can be improved in the article. The over all decision is REJECT and Re Submit.  

Thank you very much.

Author Response

Dear Reviewer,

The authors appreciate your valuable comments which help to improve the manuscript. The authors would like to highlight that this is the second round of revision for the present manuscript, and that all the valuable comments and suggestions provided on the first round by the Reviewer were much appreciated and helped significantly to improve the current version of this paper. However, some of the comments presented in this second round are completely new and were not mentioned in the first round. Nonetheless, the authors have tried to contemplate the new Reviewer’s comments as best as possible. Please find below the authors answers to Reviewer’s comments.

The authors chose to weld the copper pipes through the brazing process since it is the process that is used to weld copper pipes in the heat pump manufacturing industry. This work has been a collaboration with Bosch Thermotechnology, which is one of the strongest manufacturers of heat pumps in the market. The objective is to help the industry to improve their copper pipe welding processes and prevent product failures, contributing to sustainable manufacturing, cleaner production, and energy savings. This information has been included in the revised manuscript.

The furnace has a volume of approximately 400 litres. The temperatures were selected based on the welding temperature provided by the addition material supplier, which is 710 °C. The authors investigated temperature increase from this point to the upper limit of the melting range temperature for the addition material. The information for the addition material is provided in Anex 1.

Lines 20-22 have been reformulated. The flow of H2 in the oven is controlled by the oven itself, which has valves to control the flow of protective gas.

The authors have included more recent literature in the manuscript.

The purpose of line 145 is to alert any readers that might try performing similar studies on other materials that some materials may react with the nitrogen.

The authors kept Figures 9 and 10 as they were, in their understanding, it is easier to see the details if the two Figures are separate and with bigger dimensions.

The SEM analysis is considered important by the authors. It allows one to see the welding joints in detail and to identify several types of defects.

Despite the EDS test being a semi-quantitative test, this test was used to complement the microscopy analysis. Its purpose is to inspect the chemical compositions of the addition material and the copper pipes. Furthermore, this test was also useful to assess several surface phenomena, such as oxidation in the defect area.

Reviewer 2 Report

Comments and Suggestions for Authors

The manuscript can be accepted.

Author Response

The authors would like to thank the Reviewer for the valuable comments, which helped to improve the manuscript.

Round 3

Reviewer 1 Report

Comments and Suggestions for Authors

The authors made the necessary changes in the manuscript. The paper can be considered for publication